# Predictive Factors of Adrenal Insufficiency in Outpatients with Indeterminate Serum Cortisol Levels: A Retrospective Study

**DOI:** 10.3390/medicina56010023

**Published:** 2020-01-08

**Authors:** Worapaka Manosroi, Mattabhorn Phimphilai, Jiraporn Khorana, Pichitchai Atthakomol, Tanyong Pipanmekaporn

**Affiliations:** 1Division of Endocrinology, Department of Internal Medicine, Faculty of Medicine, Chiang Mai University, Chiang Mai 50200, Thailand; worapaka.m@gmail.com (W.M.); mphimphi@hotmail.com (M.P.); 2Clinical Epidemiology and Clinical Statistic Center, Faculty of Medicine, Chiang Mai University, Chiang Mai 50200, Thailand; jiraporn.kho@elearning.cmu.ac.th; 3Division of Pediatric Surgery, Department of Surgery, Faculty of Medicine, Chiang Mai University, Chiang Mai 50200, Thailand; 4Department of Orthopedics, Faculty of Medicine, Chiang Mai University, Chiang Mai 50200, Thailand; p.atthakomol@gmail.com; 5Department of Anesthesiology, Faculty of Medicine, Chiang Mai University, Chiang Mai, 50200 Thailand

**Keywords:** adrenal insufficiency, predictive factor, serum cortisol, ACTH

## Abstract

*Background and Objectives*: To diagnose adrenal insufficiency (AI), adrenocorticotropic hormone (ACTH) stimulation tests may need to be performed, but those tests may not be available in some institutions. In addition, they may not be necessary for some patients. The objective of this study was to identify clinical and biochemical factors that could facilitate AI diagnosis in outpatient departments and decrease the number of unnecessary dynamic tests. *Materials and Methods*: This seven-year retrospective study was performed in a tertiary care medical center. A total of 517 patients who had undergone ACTH stimulation tests in the outpatient department were identified. AI was described as a peak serum cortisol level of <18 µg/dL at 30 or 60 min after stimulation. The associations between clinical factors, biochemical factors, and AI were analyzed using the Poisson regression model and reported by the risk ratio (RR). *Results*: AI was identified in 128 patients (24.7%). Significant predictive factors for the diagnosis of AI were chronic kidney disease (RR = 2.52, *p* < 0.001), Cushingoid appearance (RR = 3.44, *p* < 0.001), nausea and/or vomiting (RR = 1.84, *p* = 0.003), fatigue (RR = 1.23, *p* < 0.001), serum basal cortisol <9 µg/dL (RR = 3.36, *p* < 0.001), serum cholesterol <150 mg/dL (RR = 1.26, *p* < 0.001), and serum sodium <135 mEq/L (RR = 1.09, *p* = 0.001). The predictive ability of the model was 83% based on the area under the curve. *Conclusion*: The easy-to-obtain clinical and biochemical factors identified may facilitate AI diagnosis and help identify patients with suspected AI. Using these factors in clinical practice may also reduce the number of nonessential dynamic tests for AI.

## 1. Introduction

Adrenal insufficiency (AI) is a rare disease which can be lethal if undiagnosed and left untreated. The most common etiology of AI is post-glucocorticoid therapy and the second most common etiology is post-pituitary surgery [1]. In Thailand, the use of herbal medicines or traditional medicines, most of which have been adulterated with glucocorticoids, has been reported [2]. Individuals taking these herbal medicines are vulnerable to steroid deficiency if the medication is stopped abruptly [3]. Common presentations of AI include weight loss, anorexia, nausea, vomiting, lethargy, and fatigue [4]. Cushingoid appearance may also be present in patients reporting a history of exogenous steroid use. According to current guidelines, the diagnosis of AI involves multiple steps. Serum morning cortisol <3 µg/dL is strongly suggestive of AI. If serum morning cortisol is greater than 18 µg/dL, AI can be ruled out. In patients with levels between 3 and 17.9 µg/dL (intermediate zone), further dynamic tests are needed [5]. The most common diagnostic test employed for these patients is the adrenocorticotropic hormone (ACTH) stimulation test, which is safe, reliable, and accurate [6].

ACTH is among the medicines which are in short supply in Thailand, where it may not be available in some institutions. As a result, some medical centers cannot perform the ACTH stimulation tests, so patients needing these tests must be referred to other centers. If clinical or biochemical factors could be identified that could help diagnose AI, particularly in patients with serum cortisol in the “intermediate” or “grey zone” range, this could help reduce unnecessary ACTH stimulation testing and conserve limited ACTH resources.

Previous studies identifying biochemical or clinical factors associated with AI have been performed in inpatient departments [7,8]. However, several confounding factors can make the interpretation of serum cortisol tests difficult, particularly in hospitalized patients. For example, serum albumin, which binds to free cortisol, can become low in acutely ill patients, leading to a false diagnosis of low serum cortisol [9]. In critically ill patients, a physiologically severe stress response can occur, causing serum cortisol to be higher than normal in the general population [10]. In contrast, serum cortisol in outpatients is less likely to be affected by such factors.

Only one retrospective study of 329 outpatients reported that a history of autoimmune disease associated with autoimmune polyendocrine syndrome, the use of glucocorticoids, fatigue symptoms, and symptoms of hypotension were predictive factors for AI [11]. It proposed that using 5.2 and 13.6 µg/dL as the lower and upper cut-off values for serum morning cortisol, respectively, rather than the standard cut-off values, could reduce the number of ACTH stimulation tests by 12%, while preserving a high sensitivity and specificity [11]. To date, there have been no studies which have incorporated both biochemical and clinical factors in a single model that can diagnose AI.

The primary objective of this study was to identify clinical and biochemical factors which could facilitate the prediction of AI in outpatients. The secondary objective was to demonstrate the accuracy of that model for AI prediction, both in a sub-group of patients with pituitary/adrenal diseases and in individuals without those diseases.

## 2. Materials and Methods

A 7-year retrospective observational cohort study to identify diagnostic prediction factors for AI was conducted using data acquired from the electronic medical charts of all patients referred to the adult endocrinology outpatient department unit at a university hospital in the northern part of Thailand during January 2010–December 2016. The study protocol has been previously reported and was approved by the Faculty of Medicine, Chiang Mai University, Ethical Committee (Ethical number: MED-2562-06193, Date of approval: 27 March 2019) [12]. The inclusion criteria were adult patients who had 0800 h serum morning cortisol between 3 and 17.9 µg/dL (83–500 nmol/L) and who had undergone ACTH stimulation tests. Exclusion criteria were patients with incomplete results from ACTH stimulation testing, females currently taking oral contraceptive pills containing estrogen, patients who had had pituitary tumor surgery within the previous 2 months, and patients suspected of having congenital adrenal hyperplasia.

### 2.1. ACTH Stimulation Test Protocol

The ACTH stimulation test protocol has been defined in detail elsewhere [12]. In brief, patients receiving glucocorticoids were instructed to stop taking those substances at least 24 h before testing. For each patient, either low-dose (1 µg) ACTH stimulation tests (LDT) or high-dose (250 µg) ACTH stimulation tests (HDT) were performed. Due to the ACTH shortage in Thailand in the period of May 2010–March 2014, only LDT were conducted during that period, while HDT were conducted for stimulation testing during April 2014–December 2016. The ACTH stimulation tests were usually conducted between 1000 and 1300 h. The total serum cortisol level was drawn at 0 (basal cortisol), 30, and 60 min following the intravenous administration of either 1 µg or 250 µg ACTH. The 1 µg ACTH was prepared by the hospital pharmacists by mean of diluting a 250 µg ampule of ACTH with normal saline, which was then transferred to 1 mL syringes and stored at 2–8 °C. Solutions were used within 60 days of preparation.

### 2.2. Data Collection

The demographics, underlying history, and clinical data for patients were retrieved from medical records of patients’ visits for ACTH stimulation tests. All biochemical data were collected within the 3 months prior to or following the tests. Data on the serum albumin, cholesterol, and creatinine obtained during acute illness episodes or within three months of admission were excluded. The measurement of serum cortisol levels was performed by an electrochemiluminescence immunoassay (ECLIA) (Elecsys^®^ Cortisol 1010, Roche Diagnostics, Laval, QC, Canada). The variation of intra- and inter-assay coefficients for serum cortisol was less than 10%.

### 2.3. Definitions

The serum morning cortisol level was determined from serum cortisol samples drawn at 0800 h. Serum basal cortisol was acquired from serum cortisol samples drawn between 0900 and 1300 h and prior to ACTH administration. A peak serum cortisol level of <18 µg/dL (<500 nmol/L) at 30 or 60 min after either LDT or HDT was described as having AI [13]. The definition of history of glucocorticoid use was the use of glucocorticoids, including herbal or traditional medicines speculated as exhibiting adulteration with glucocorticoids, which were not prescribed by physicians during a period of at least 3 weeks prior to ACTH stimulation testing. A history of pituitary surgery was defined as having undergone tumor removal (except corticotropin-producing tumors) at the pituitary or sellar/suprasellar lesions extending to the pituitary gland. A pituitary tumor was defined as a sellar/suprasellar tumor extending to the pituitary gland with no history of tumor removal. Hormonal deficiencies were defined as any pituitary hormonal deficiency (secondary hypothyroidism, secondary hypogonadism, growth hormone deficiency, or diabetes insipidus) from any cause, apart from post-pituitary surgery or a pituitary tumor (e.g., infiltrative disease, transcription factor defect, lymphocytic hypophysitis, post radiation therapy, and empty sella syndrome). Adrenal disease was defined as a history of adrenal gland surgery, Cushing’s syndrome following adrenalectomy, adrenal gland infection, an adrenal gland tumor, or autoimmune disease of the adrenal gland. Symptoms of AI included fatigue, syncope, weight loss, intractable nausea and vomiting, and orthostatic hypotension. Weight loss was defined as a loss of 5% of body weight in one month or 10% over a period of six months or longer [14]. Cushingoid appearance was defined as the physician documentation of at least one sign of excess glucocorticoid, e.g., moon face, facial plethora, easy bruising, and hirsutism. Chronic kidney disease (CKD) was defined as an estimated glomerular filtration rate (eGFR) of less than 30 mL/min/1.73 m^2^, as determined by MDRD (modification of diet in renal disease) formula. Eosinophilia was defined as total eosinophil levels >1500/µL of blood. Lymphocytosis was defined as total lymphocytes >4000/µL of blood. Primary AI was defined as patients with any history of adrenal gland diseases (e.g., Addison’s disease, infection, and infiltrative diseases of the adrenal gland) or post-adrenal gland surgery. The other causes were defined as secondary AI.

Subgroup analyses were conducted for patients with pituitary/adrenal diseases, including those with a history of pituitary surgery, a pituitary tumor, or pituitary hormonal deficiencies and for patients without pituitary/adrenal diseases, but who had a history of exogenous steroid use, AI symptoms, or other signs indicating the need to test for AI.

### 2.4. Statistical Analysis

The data were analyzed using STATA program 15.1. The statistical significance level was described as two-tailed with a *p*-value < 0.05. To determine the distribution of the data, the Shapiro–Wilk test was used. For categorical variables, counts or percentages are reported; for normally distributed continuous variables, means and standard deviation (SD) are presented. For non-normally distributed continuous variables, medians with interquartile ranges are presented. For continuous data, the univariable analysis was analyzed with the independent *t*-test for normally distributed variables and the Wilcoxon rank-sum test was employed for non-normally distributed variables. For categorical variables, univariable analysis was conducted by the Fisher exact test. Additionally, univariable and multivariable analyses of the diagnostic prediction factors for AI were performed using Poisson regression and are reported as the risk ratio (RR) with a 95% confidence interval (CI). The prediction model was developed using stepwise selection, removing variables with *p*-value > 0.2 by backward elimination. To assess the performance of the model, the area under the receiver operating characteristic (AuROC) curve was plotted. Variables with missing data >5% were dealt with using multiple imputation regression. The calculated sample size was based on a report by Perton et al. [11]. A sample size of at least 430 patients was estimated to give 80% power at the 5% significance level (two-sided), with an odds ratio of 0.5 of detecting AI for a specific risk factor [11].

## 3. Results

### 3.1. Baseline Characteristics and Univariable Analysis of Factors Associated with AI

The overall study flow is shown in Figure 1. Of the 517 outpatients (252 females and 265 males) who had undergone ACTH stimulation testing, 24.75% (128/517) had documented AI. Their mean age was 50.74 ± 17.55 years; the mean age was significantly higher in the AI group than in the non-AI group. Of all the outpatients, 33.07% (171/517) had symptoms indicative of AI, the most common indication for ACTH stimulation testing. Baseline characteristics are shown in Table 1. The second most common indication for testing was pituitary hormonal deficiencies, with a prevalence of 29.21% (151/517), followed by a history of exogenous glucocorticoid use, with a value of 25.53% (132/517). Cushingoid appearance was found in 50% (66/132) of those with a history of exogenous glucocorticoid use. Demographic data categorized by subgroups of those with and without pituitary/adrenal diseases and those with primary and secondary AI are provided in the Appendix A.

In the univariable model, the significant clinical predictive factors for AI were age, a history of hypertension, CKD, patient-reported fatigue, Cushingoid appearance, a history of exogenous steroid use (particularly among those who reported using herbal and/or traditional medicine), patients with adrenal diseases, and those with hyponatremia. Among the biochemical factors, serum morning cortisol, serum basal cortisol, and serum albumin were the factors significantly related to AI. The *p*-value and RR for each parameter are shown in Table 1 and Table 2.

### 3.2. Multivariable Analysis for Clinical and Biochemical Factors Related to AI

Only basal cortisol was included in the multivariable model based on the study from our group, which showed that serum basal cortisol demonstrated a higher diagnostic accuracy than serum morning cortisol [12]. Following multivariable analysis, the significant clinical predictive factors used in the stimulation tests for AI clustered by ACTH dose were CKD (RR = 2.52, *p* < 0.001), Cushingoid appearance in those with exogenous steroid and/or herbal medicine ingestion (RR = 3.44, *p* < 0.001), symptoms of nausea and/or vomiting (RR = 1.84, *p* = 0.003), and symptoms of fatigue (RR = 1.23, *p* < 0.001). The biochemical predictive factors for AI were serum basal cortisol < 9 µg/dL (RR = 3.36, *p* < 0.001), serum cholesterol < 150 mg/dL (RR = 1.26, *p* < 0.001) and serum sodium < 135 mEq/L (RR = 1.09, *p* = 0.001) (Table 3). The ROC curve for the model assessing the predictive performance incorporating all significant predictive factors had an AuROC of 0.83 (0.80–0.87) (Figure 2).

### 3.3. Subgroup Analysis of Patients with and without Pituitary/Adrenal Diseases

When the multivariable model described above was used with the subgroups that had pituitary/adrenal diseases and those that did not, all the factors showed a *p*-value of <0.05. The AuROC in the subgroup with pituitary/adrenal diseases was 0.80 (0.76–0.84), while in the subgroup without pituitary/adrenal diseases, the AuROC was 0.87 (0.83–0.94). Data are shown in the Appendix A.

## 4. Discussion

An important finding of this study is that various clinical and biochemical factors can help diagnose AI. The final model incorporating these factors yielded a high diagnostic value of 83% based on AuROC.

Compared to a prior study [11], our study did not find a significant association between AI and symptoms of hypotension. The present study did find that CKD is one of the factors associated with AI, although the details of the relationship between CKD and AI remain controversial. Previous studies reported that the serum half-life of cortisol is prolonged in CKD. Additionally, both elevated and normal cortisol levels in CKD patients have been demonstrated [15,16,17,18]. Multiple studies have reported a normal adrenal function in CKD patients [19,20]. The present study found that patients with CKD had a risk of AI of 2.52 times compared to patients without CKD. This finding indicates that physiological changes in cortisol metabolism in CKD were not sufficient to explain the elevated risk of AI found in our study. Further analysis showed that patients with CKD had a higher mean age than those without CKD (60.06 ± 16.28 versus 50.44 ± 17.52 years, *p* = 0.031). This suggests that patients with CKD are more likely to have multiple age-related comorbidities, which can aggravate AI. For example, a study of hemodialysis patients indicated that AI resulted from various causes, e.g., tuberculosis, amyloidosis, and steroid withdrawal [21]. However, our study did not find an association between those conditions and CKD. Further research on the relationship between AI and CKD is warranted. Physicians should be aware that AI may occur concurrently with CKD and that further investigations may be required to confirm a diagnosis of AI.

As hypothesized, Cushingoid appearance in cases of exogenous steroid ingestion was demonstrated to be significantly linked to a nearly four-fold increase in the risk of AI. A previous meta-analysis reported that AI can occur in 6.8%–60% of patients who use corticosteroids [22]. Similarly, in our study, half the patients who used exogenous steroids were documented as having AI. Cushingoid appearance in individuals with AI who also use corticosteroids has been described in multiple case reports of topical steroid therapy [23,24]. High-dose or chronic glucocorticoid use from prescribed or surreptitious use can result in Cushingoid appearance. Prolonged suppression of the hypothalamic-pituitary-adrenal (HPA) axis from long-term glucocorticoid use can result in secondary AI after tapering or abrupt discontinuation of the glucocorticoid dose. For those reasons, AI should be suspected in patients with Cushingoid characteristics and further evaluation should be considered. However, mischaracterization of Cushingoid appearance by physicians can easily occur as this characteristic is a subjective finding. Moreover, the reversible nature of this manifestation may mislead the diagnosis. Therefore, in practice, Cushingoid appearance as a predictor of AI should be evaluated in conjunction with other predictive factors to increase the diagnostic accuracy.

Symptoms of AI are usually non-specific, which often results in a delayed diagnosis. The present study found that symptoms of nausea and/or vomiting and fatigue are predictive of AI. The incidence of nausea and/or vomiting and fatigue in AI patients in studies has been reported in approximately 49%–62% and 84%–95% of cases, respectively [25]. Other symptoms, including weight loss and orthostatic hypotension, have also been shown to be common presenting symptoms in AI patients. However, in this study, no significant association was found between either weight loss or orthostatic hypotension and AI. As nausea and/or vomiting and fatigue are non-specific and can be indicative of diseases other than AI, the use of nausea/vomiting and fatigue symptoms alone to diagnose AI may not be appropriate; diagnosis should incorporate other factors to increase the predictive accuracy.

Among the biochemical factors studied, cholesterol < 150 mg/dL was the strongest predictive factor for AI, with an RR of 1.26. Previous studies have reported an association between serum cholesterol and AI in cirrhotic patients. Spadaro et al. stated that low total cholesterol exhibited an association with AI, while Park et al. found no association between serum cholesterol and AI in cirrhotic patients [26,27]. As cholesterol is a precursor of glucocorticoid synthesis, AI may be present when serum cholesterol is lower than normal. An animal study demonstrated that in mice treated with high-dose simvastatin for 3 days, AI could be aggravated by their hypocholesterolemic condition [28]. Our results appear to support that hypothesis. Serum Na < 135 mEq/L has been found to be another predictive factor for AI, with a risk ratio of 1.09. This association could be explained by the hypersecretion of ADH observed in cortisol deficiency patients [29] as ADH hypersecretion can cause water retention and a reduction in the plasma sodium concentration [30].

As hypothesized, the present study found that if serum basal cortisol is <9 µg/dL, the likelihood of a patient having AI increases by more than three times. The cut-off level of basal cortisol was derived from the median values of basal cortisol in our cohort. The proposed cut-off level to rule-in AI identified in this study is higher than in previous studies [11,31]. If this higher cut-off level were used in combination with other factors, it could reduce the number of ACTH stimulations performed. Presently, clinical practice guidelines recommend that in institutions where ACTH stimulation testing is not available, serum morning cortisol (between 6 A.M. and 9 A.M.), together with the ACTH level, can be used to diagnose AI; there is no published guideline supporting the use of serum random or basal cortisol to diagnose AI [13]. In the previous study from our group, we found that serum basal cortisol demonstrated a better diagnostic performance than serum morning cortisol, and for that reason, only serum basal cortisol was employed in the model [12]. That decision is in accord with a previous study which found that basal cortisol had a high diagnostic accuracy for AI, with an AuROC of 0.88 [32]. This suggests that basal cortisol may be used as one of the predictive factors for AI where no data on serum morning cortisol is available. Further study on the use of basal cortisol in combination with other clinical factors to develop an easy-to-use scoring system to predict AI is warranted.

When categorized into two subgroups—those with and without pituitary/adrenal diseases—the final model demonstrated a good accuracy in AI prediction, with an AuROC > 0.80 in both subgroups. The reason for classifying the population into these subgroups was that medical data of patients with a history of pituitary or adrenal diseases were objective and have been reported to be significantly linked to AI [33,34,35,36,37]. These data were measurable and observable, unlike another subgroups, which included reported information on symptoms of AI or a history of use of glucocorticoids or traditional medicines suspected of being contaminated with glucocorticoids. Those reported histories were fairly subjective. Additionally, symptoms of AI can mimic other diseases [38]. The final model proposed in this study could potentially be generalized to an entire population, including both patients with and without pituitary/adrenal diseases. The large sample size in this study provided an adequate power of analysis; however, these findings must be considered preliminary as the sample size for each subgroup was not specified beforehand.

The present study demonstrates that a statistically significant association exists among multiple clinical and biochemical factors and AI. The relationship suggests that in institutions where ACTH stimulation testing is not available, patients with these clinical symptoms and biochemical factors should be referred to another institution for further dynamic tests for AI. A practical risk scoring system incorporating these characteristic symptoms and factors needs to be developed. The relationships among the predictive factors identified in this study could be clarified by the underpinning physiology and the supporting evidence discussed in this study. Furthermore, the finding regarding basal cortisol for AI was demonstrated. The present study included adjustment for potentially confounding factors and used cluster analysis to minimize the effect of variation in the ACTH dosages used in the ACTH stimulation tests. A high number of sample sizes gave an adequate power of analysis. The various indications of the need for ACTH stimulation testing are broadly applicable to actual clinical practice.

There are some limitations of this study. First, the retrospective nature of the study and the fact that it was a single-center study may limit the generalizability of the results and their applicability to other institutions. Second, the study included a diverse population, so the results may not be valid for some subgroups. Conversely, that diversity would suggest that the results could be applicable for a variety of outpatient populations. Among the biochemical factors reported to be related to AI, serum CRP (C-reactive protein) and calcium levels were not investigated in our cohort. The patients with overt AI were not included in this study as only those with serum morning cortisol levels above 3 µg/dL were included in our cohort. Another limitation is that the study did not explicitly investigate whether the diagnosed AI was a primary or secondary cause by using specific laboratory tests, e.g., serum ACTH and DHEA-S (dehydroepiandrosterone sulfate) levels. However, this can be assumed from the indication of the tests, clinical characteristics, and imaging studies. Additional prospective research regarding this issue may be needed.

## 5. Conclusions

Serum cortisol, cholesterol, sodium, and a number of other easy-to-assess clinical factors can help predict the occurrence of AI in outpatients. These predictive factors can potentially allow physicians, particularly those in areas where dynamic tests for diagnosing AI are not readily available, to identify patients with indeterminate serum morning cortisol levels and refer them to other institutions for further testing. The predictive factors identified in this study could also result in time- and cost-savings for medical facilities providing ACTH stimulation tests as the number of tests could be reduced if these factors can help identify patients with a low risk of AI. Further research is needed to transform these predictive factors into applied clinical risk scores.

## Figures and Tables

**Figure 1 medicina-56-00023-f001:**
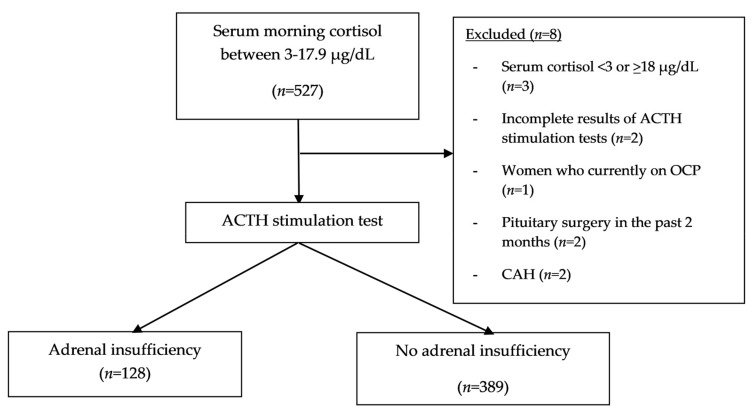
Study flow.

**Figure 2 medicina-56-00023-f002:**
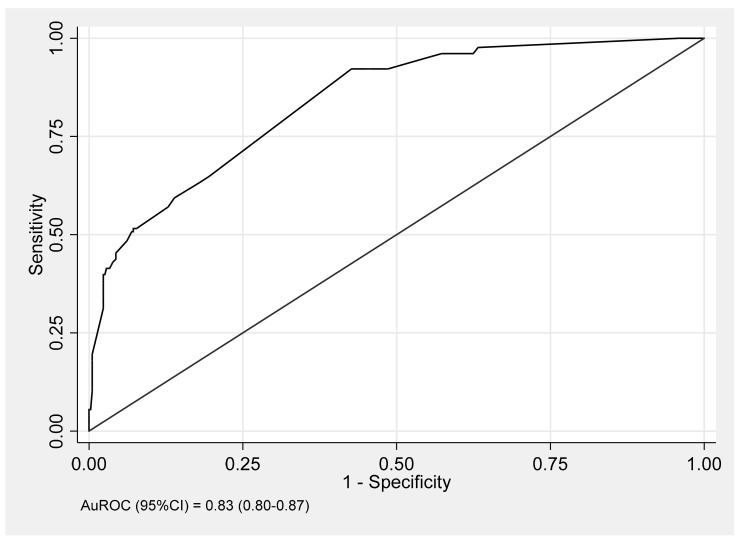
Area under the receiver operating characteristic (AuROC) of the predictive model for adrenal insufficiency incorporating clinical and biochemical factors predicted by the predictive factors (curved line) and a 50% chance of prediction (diagonal line).

**Table 1 medicina-56-00023-t001:** Baseline characteristics of patients who had undergone adrenocorticotropic hormone (ACTH) stimulation tests with adrenal insufficiency (*n =* 128) and no adrenal insufficiency (*n =* 389).

Characteristic	Adrenal Insufficiency	No Adrenal Insufficiency	RR *	95% CI *	*p*-Value
	(*n =* 128)	(*n =* 389)			
Age, (mean ± SD) (yr)	56.61 ± 16.97	48.80 ± 17.32			<0.001
Age group					
- <50 years old, n (%)	40 (31.25)	174 (44.73)	(Ref)		
- ≥50 years old, n (%)	88 (68.75)	215 (55.27)	1.55	1.06–2.25	0.024
Male, n (%)	68 (53.13)	197 (50.64)	1.07	0.87–1.68	0.673
Weight, (mean ± SD) (kgs)	58.23 ± 13.02	61.41 ± 16.07	0.98	0.97–1.00	0.083
BMI, (mean ± SD) (kg/m^2^)	22.87 ± 5.30	24.14 ± 6.65	0.97	0.96–1.00	0.065
ACTH stimulation dose, n (%)					
- 1 µg	41 (32.03)	150 (38.56)	(Ref)		
- 250 µg	87 (67.97)	239 (61.44)	1.24	0.85–1.80	0.25
Systolic blood pressure, (mean ± SD) (mmHg)	121.37 ± 21.96	121.40 ± 9.93	0.99	0.99–1.00	0.99
Diastolic blood pressure, (mean ± SD) (mmHg)	71.99 ± 15.46	73.74 ± 12.84	0.99	0.98–1.00	0.271
Underlying disease, n (%)					
- Diabetes mellitus	21 (16.41)	57 (14.69)	1.1	0.69–1.75	0.684
- Hypertension	41 (32.03)	83 (21.39)	1.48	1.11–2.69	0.035
- Chronic kidney disease	11 (8.59)	5 (1.29)	2.94	1.58–5.46	0.001
- Autoimmune disease	24 (18.75)	55 (14.14)	1.27	0.82–1.99	0.271
- Cancer	4 (3.13)	12 (3.08)	1.01	0.37–2.73	0.984
Symptom, n (%)					
- Fatigue	36 (28.13)	71 (18.25)	1.49	1.01–2.20	0.039
- Weight loss	4 (3.13)	22 (5.66)	0.6	0.22–1.64	0.329
- Orthostatic hypotension	14 (10.94)	36 (9.25)	1.14	0.65–1.99	0.628
- Nausea/vomiting	4 (3.13)	5 (1.29)	1.82	0.67–4.92	0.238
Indication for ACTH testing, n (%)					
- Exogenous steroid use	59 (46.09)	73 (18.77)	2.49	1.76–3.53	<0.001
- Post-surgery of pituitary	19 (14.84)	86 (22.11)	0.68	0.42–1.11	0.127
- Pituitary tumor	20 (17.09)	97 (82.91)	0.63	0.39–1.02	0.06
- Pituitary hormonal deficiencies	32 (25.0)	119 (30.59)	0.8	0.54–1.20	0.296
- Adrenal disease	11 (8.59)	7 (1.80)	2.6	1.40–4.83	0.002
- Symptoms of adrenal insufficiency	40 (31.25)	131 (33.68)	0.91	0.63–1.33	0.666
- Other indications					
- Hyponatremia	9 (22.5)	9 (6.87)	2.22	1.16–4.23	0.015
- Hypoglycemia	2 (5.00)	12 (9.16)	0.57	0.14–2.30	0.431
Cushingoid appearance in exogenous steroid use	51 (39.8)	15 (3.86)	4.52	3.17–6.44	<0.001
Dose of glucocorticoids equivalent to prednisolone, n (%)					
- Unknown	28 (47.46)	30 (41.10)	(Ref)		
- 0.5–5.0 mg	23 (38.98)	32 (43.84)	1.9	1.21–3.00	0.005
- >5.0–20 mg	7 (11.86)	9 (12.33)	1.99	0.92–4.30	0.077
- >20 mg	1 (1.69)	2 (2.74)	1.52	0.21–10.91	0.076
Type of steroid use, n (%)					
- Prednisolone	18 (47.37)	25 (54.35)	1.58	0.93–2.65	0.085
- Dexamethasone	3 (11.54)	2 (7.14)	2.23	0.70–7.10	0.172
- Triamcinolone	1 (1.69)	5 (6.85)	0.67	0.09–4.79	0.691
- Traditional medicine or herbal use	36 (76.60)	41 (82.00)	1.87	1.23–2.83	0.003
History of pituitary surgery, n (%)					
- Microadenoma	3 (15.79)	9 (10.47)	(Ref)		
- Macroadenoma	16 (84.21)	77 (89.53)	1.09	0.45–2.64	0.832
Other hormonal deficiencies, n (%)					
- Gonadotropin	10 (41.67)	45 (55.56)	0.61	0.31–1.18	0.148
- Thyroid	26 (96.30)	1 (3.70)	0.92	0.58–1.47	0.746
- Growth hormone	4 (20.00)	12 (20.34)	0.82	0.30–2.26	0.714
- Diabetes insipidus	5 (27.78)	29 (42.03)	0.5	0.20–1.24	0.139

* Univariable analysis by Poisson regression.

**Table 2 medicina-56-00023-t002:** Baseline biochemical investigations of patients who had undergone ACTH stimulation tests with adrenal insufficiency (*n =* 128) and no adrenal insufficiency (*n =* 389).

Characteristic	Adrenal Insufficiency	No Adrenal Insufficiency	RR *	95% CI *	*p*-Value
	(*n =* 128)	(*n =* 389)			
Serum morning cortisol (mean ± SD) (µg/dL)	7.75 ± 2.78	9.55 ± 3.31			<0.001
<9 µg/dL, n (%)	84 (65.53)	188 (48.33)	(Ref)		
≥9 µg/dL, n (%)	44 (34.38)	201 (51.67)	1.71	1.19–2.47	0.004
Serum basal cortisol (mean ±SD) (µg/dL)	6.15 ± 3.14	10.80 ± 0.26			<0.001
<9 µg/dL, n (%)	104 (81.25)	165 (42.42)	(Ref)		
≥9 µg/dL, n (%)	24 (18.75)	224 (57.58)	0.25	0.16–0.39	<0.001
Serum potassium (mean ± SD) (mEq/L)	3.98 ± 0.58	4.05 ± 0.47			0.188
<3 mEq/L, n (%)	2 (1.56)	4 (1.03)	(Ref)		
≥3 mEq/L, n (%)	126 (98.44)	385 (98.97)	0.99	0.83–1.18	0.99
Serum sodium (mean ± SD) (mEq/L)	138.51 ± 5.54	138.17 ± 12.45			0.761
<135 mEq/L, n (%)	20 (15.63)	55 (14.14)	(Ref)		
≥135 mEq/L, n (%)	108 (84.38)	334 (85.86)	0.91	0.56–1.47	0.72
Eosinophilia, n (%)	11 (10.58)	93 (89.42)	0.91	0.48–1.70	0.768
Lymphocytosis, n (%)	66 (63.46)	223 (75.08)	0.83	0.59–1.18	0.324
Serum albumin (mean ± SD) (g/dL)	3.73 ± 0.72	3.97 ± 0.55			<0.001
<3 g/dL	20 (15.63)	108 (84.38)	(Ref)		
≥3 g/dL	17 (4.37)	372 (95.63)	0.41	0.25–0.67	<0.001
Total cholesterol (mean ± SD) (mg/dL)	177.86 ± 51.17	183.75 ± 50.37			0.252
<150 mg/dL	39 (30.47)	81 (20.82)	(Ref)		
≥150 mg/dL	81 (20.82)	308 (19.18)	0.68	0.47–1.00	0.053

* Univariable analysis by Poisson regression.

**Table 3 medicina-56-00023-t003:** Multivariable risk regression of predictive factors for adrenal insufficiency clustered by ACTH dose.

Factor	RR	95% CI	*p*-Value
Chronic kidney disease	2.52	2.02–3.14	<0.001
Cushingoid appearance in exogenous steroid and/or herbal medicine use	3.44	2.16–5.47	<0.001
Nausea and/or vomiting	1.84	1.24–2.75	0.003
Fatigue	1.23	1.11–1.37	<0.001
Serum basal cortisol < 9 µg/dL	3.36	3.23–3.49	<0.001
Cholesterol < 150 mg/dL	1.26	1.20–1.32	<0.001
Serum sodium < 135 mEq/L	1.09	1.04–1.15	0.001

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
