# Peer review of "Predictive Factors of Adrenal Insufficiency in Outpatients with Indeterminate Serum Cortisol Levels: A Retrospective Study"

_1010-660X, 2020, doi:10.3390/medicina56010023_

Round 1

Reviewer 1 Report

The authors tried to identify clinical and biochemical factors that could facilitate

adrenal insufficiency (AI) diagnosis in outpatient departments and decrease the number of unnecessary dynamic tests. They performed seven-year retrospective study was performed in a tertiary care medical center. They found that AI was diagnosed in 128 patients (24.7%). The significant predictive factors for diagnosis of AI were chronic kidney disease (RR=2.52, p<0.001), Cushingoid appearance (RR=3.44, p<0.001), nausea and/or vomiting (RR=1.84, p=0.003), fatigue (RR=1.23, p<0.001), serum basal cortisol <9 μg/dL (RR=3.36, p<0.001), serum cholesterol <150 mg/dL (RR=1.26, p<0.001) and serum sodium <135 mEq/L (RR=1.09, p=0.001). The predictive ability of the model was 83% based on area under the curve. They concluded that the easy-to-obtain clinical and biochemical factors identified may facilitate AI diagnosis and help identify patients with suspected AI. Using these factors in clinical practice may also reduce the number of nonessential dynamic tests

Those data are very interesting and also informative, while there are a couple of major problems as described below.

Major problems:

The author should clearly describe primary or secondary AI for analyzing in this study. The author need to analyze by more factors relating AI, such as CRP, blood glucose level, BMI, blood pressure, serum Ca concentration and so on. What do you think the importance of Cushingoid appearance for speculating AI? You need to describe ACTH and DHEA-S levels in all patients.

Minor point

The author needs to explain how to differentiate overt AI from latent or subclinical AI.

Reviewer 2 Report

This manuscript titled as, ‘Predictive Factors of Adrenal Insufficiency in outpatients with Indeterminate Serum Cortisol levels: A retrospective study’ has novel data that may be considered for publication.

However, the following suggestions and comments need to be addressed satisfactorily before acceptance.

Specific comments:

Title: good

Abstract: Overall, it is good.

However, the following sentence needs to be rewritten

“To diagnose adrenal insufficiency (AI), adrenocorticotropic 17

 hormone (ACTH) stimulation tests may need to be performed, but that test may not be available in 18

 some institutions and is not necessary in some patients”.

Key words: ‘ACTH’ may have to be added to the list.

Introduction:

1.Rewrite the following sentence:

‘The most common etiology of AI is post-glucocorticoid therapy followed by post-pituitary 39

 Surgery’

Rewrite the following sentence:

In Thailand, use of what are commonly called “herbal medicines” or “traditional 40

 medicines,” most of which have been adulterated with glucocorticoids, has been reported

Rewrite the following sentence by splitting it into two sentences:

If clinical or 52

 biochemical factors could be identified that could help diagnose AI, particularly in patients with

 serum cortisol in the “intermediate” or “grey zone” range, it could help reduce unnecessary ACTH

 stimulation testing and help conserve limited ACTH resources

Rewrite the following sentence:

Most studies to identify biochemical or clinical factors which could facilitate AI diagnosis have 56

 been performed in inpatient department

Materials and methods: Good

Results: Good

The supplementary data and figures are good. They can be included in the results section and discussion section may add few small paragraphs for these data as well.

1.Rewrite the following sentence:

‘An important finding of present study is that various clinical and biochemical factors can 207

 facilitate AI diagnosis’.

Rewrite the following sentence:

‘The serum half-life of cortisol is prolonged in CKD, with both elevated and normal cortisol levels having been reported’

Rewrite the following sentence:

The present study found that patients with CKD had a risk of AI 2.52 times that of patients without CKD.

Rewrite the following sentence:

That indicates physiological changes in cortisol metabolism in CKD per se were not sufficient to explain the elevated risk of AI found in our study.

Rewrite the following sentence:

However, mischaracterization of Cushingoid appearance can easily occur as the diagnosis of this 232

 characteristic is quite subjective

Round 2

Reviewer 1 Report

Overall descriptions are well revised.